**Data Availability Statement:** The minimal relevant data are within the paper and its Supporting Information files. Moreover, S1 MT_Small_Dataset

# Automatic semantic segmentation of breast tumors in ultrasound images based on combining fuzzy logic and deep learning—A feasibility study

**Samir M. Badawy**[1☯], **Abd El-Naser A. Mohamed**[2☯], **Alaa A. Hefnawy**[3☯], **Hassan E. Zidan**[3☯], **Mohammed T. GadAllah**[3☯]*, **Ghada M. El-Banby**[1☯]

1 Industrial Electronics and Control Engineering Department, Faculty of Electronic Engineering, Menoufia University, Menoufia, Egypt, 2 Electronics and Electrical Communications Engineering Department, Faculty of Electronic Engineering, Menoufia University, Menoufia, Egypt, 3 Computers and Systems Department, Electronics Research Institute (ERI), Cairo, Egypt

☯ These authors contributed equally to this work.
* mohammed.tag.1986@eri.sci.eg, mohammed.tag.1986@gmail.com, mohamed_msc_1986@yahoo.com

## Abstract

Computer aided diagnosis (**CAD**) of biomedical images assists physicians for a fast facilitated tissue characterization. A scheme based on combining fuzzy logic (**FL**) and deep learning (**DL**) for automatic semantic segmentation (**SS**) of tumors in breast ultrasound (**BUS**) images is proposed. The proposed scheme consists of two steps: the first is a FL based preprocessing, and the second is a Convolutional neural network (**CNN**) based SS. Eight well-known CNN based SS models have been utilized in the study. Studying the scheme was by a dataset of 400 cancerous BUS images and their corresponding 400 ground truth images. SS process has been applied in two modes: **batch** and **one by one** image processing. Three quantitative performance evaluation metrics have been utilized: global accuracy (**GA**), mean Jaccard Index (mean intersection over union (**IoU**)), and mean BF (Boundary F1) Score. **In the batch processing mode**: quantitative metrics' average results over the eight utilized CNNs based SS models over the 400 cancerous BUS images were: **95.45**% GA instead of **86.08**% without applying fuzzy preprocessing step, **78.70**% mean IoU instead of **49.61**%, and **68.08**% mean BF score instead of **42.63**%. Moreover, the resulted segmented images could show tumors' regions more accurate than with only CNN based SS. **While, in one by one image processing mode:** there has been no enhancement neither qualitatively nor quantitatively. So, only when a batch processing is needed, utilizing the proposed scheme may be helpful in enhancing automatic ss of tumors in BUS images. Otherwise applying the proposed approach on a one-by-one image mode will disrupt segmentation's efficiency. The proposed batch processing scheme may be generalized for an enhanced CNN based SS of a targeted region of interest (**ROI**) in any batch of digital images. A modified small dataset is available: https://www.kaggle.com/mohammedtgadallah/mt-small-dataset (S1 Data).

are also available on https://www.kaggle.com/mohammedtgadallah/mt-small-dataset.

**Funding:** The authors received no specific funding for this work.

**Competing interests:** The authors have declared that no competing interests exist.

## Introduction

Breast cancer is regarded as the second common cancer globally after lung cancer, the fifth common reason for cancer death [1]. Efficient screening of breast cancer is important because typically this cancer has no symptoms [2]. X-ray has been utilized for early diagnosis of breast cancer starting in 1980s by screening breast (a mammography scan) producing output image called a mammogram [1]. Mammography scan causes a painful breast compression, exposure to ionizing radiation, non-optimal sensitivity and specificity, and poor sensitivity of cancer detection in dense breasts [3, 4]. BUS has the potential to be utilized as mammography's adjunct [5]. A BUS's improvement is the automated BUS (**ABUS**), having the ability in decreasing operator's dependence when compared to conventional handheld ultrasound scans [6].

Artificial intelligence (AI) including DL has emerged recently though various applications in healthcare [7–9]. Efficient cancer characterization in BUS images can be obtained by appropriate automatic SS scheme [10–16]. Efficient DL based automatic SS, being a challenging task, is aiming to label each pixel in an image with a corresponding class using supervised learning [11, 17–22]. For BUS images, SS is the classification task for two tissue's classes: normal and abnormal [10, 11, 14, 16, 23, 24]. Image preprocessing enhancement before automatic SS could play an important role in achieving more accurate and efficient segmented image. Fuzzy—based image processing has been utilized throw literature achieving more than one success in the field of image enhancement [25–30].

In this paper, we introduce an automatic SS approach for batch processing by combining a Fuzzy method for contrast enhancement using an intensification operator as a preprocessing enhancement step before starting a known CNN based SS scheme. Eight CNN based SS schemes has been applied from [11]: FCN with AlexNet network, UNet network, SegNet using VGG16, SegNet using VGG19, DeepLabV3+ using ResNet18, DeepLabV3+ using ResNet50, DeepLabV3+ using MobileNet-V2, and DeepLabV3+ using Xception networks. The output segmentation results has been evaluated by three measures: global accuracy, mean IoU also called mean Jaccard Index, and mean BF (Boundary F1) Score. The proposed approach has been applied using a 400 BUS's images and their 400 ground truth images taken from [31] in two modes: batch processing and one by one image processing. A modest dataset named **MT_Small_Dataset** (based on the 800 images taken from [31]) has been adjusted and arranged for 1200 images: 400 adjusted to size 128 by 128 by 3 and the same 400 (size 128 by 128 by 3) after applying fuzzy based contrast enhancement and 400 image ground truth adjusted in gray level [0 255], size (128 by 128), and have two classes "1" represents normal tissue and "2" represents cancerous tissue to be appropriate for evaluating the most known CNN based semantic segmentation output images using MATLAB [32].

The rest of the presented paper organized as follows: section II: related work through recent few years. Section III: covers the materials used and the methods applied in our study. Section IV: displays and discusses the obtained quantitative results and also displays samples for qualitative results. Finally, section V: the conclusion.

## Related work

Q. Huang et al [10], have introduced semantic classification of superpixels for BUS image's segmentation as follow: cropping an ROI in the base image by a selection of two diagonal points, then histogram equalization, bilateral filter and pyramid mean shift filter are applied for enhancing the image, dividing the cropped image into many superpixels by simple linear iterative clustering (SLIC), and finally the classification process by a back propagation neural network (BPNN) followed by k-nearest neighbor (KNN) achieving the final result.

W. Gomez and W. Pereira [11], have introduced their comparative study for SS of breast tumors in ultrasound images utilizing eight well established public convolutional neural networks (**CNNs**): FCN with AlexNet network, UNet, SegNet using VGG16 and VGG19, and DeepLabV3+ using ResNet18, ResNet50, MobileNet-V2, and Xception. They have achieved their study aiming to select an efficient CNN-based segmentation model to be further utilized in CAD systems. Their study has been done by applying transfer learning (TL) for fine tuning these eight CNNs to segment BUS images into two classes, normal and cancerous pixels, using more than 3000 BUS images (brought from seven ultrasound machine models) for training and validation. From the final performance evaluation of their study, they have recommended using ResNet18 when trying to implement a fully automated end-to-end CAD system. Moreover, they have made the eight generated CNN models in their study available to all researchers throw a link mentioned in their paper.

K. Huang et al [12], have introduced their study of fuzzy SS of BUS image with breast anatomy constraints, by two steps: first, fuzzy FCN for good segmentation, and second, using breast anatomy constrained conditional random fields to fine-tune the segmentation result.

Yuan Xu et al [13], have introduced their machine learning based work of medical BUS images' segmentation, proposing a CNNs based fully automatic BUS images' segmentation method into four major tissues: skin, fibroglandular tissue, mass, and fatty tissue, resulting in efficient automated segmentation providing a helpful reference to radiologists for better breast cancer characterization and breast density assessments.

K. Huang et al [14], have introduced medical knowledge constrained SS for BUS images, proposing an approach using information extended images for training an FCN for SS of BUS images into three classes: cancer, mammary layer, and background, followed by applying layer structure information, locating breast cancers into the mammary layer, conducting breast cancer segmentation by a conditional random field (CRF) producing more precise segmentation result.

Y. Lei et al [15], have introduced their study for breast tumor segmentation in three dimensional (3D) ABUS, proposing a developed Mask scoring region-based CNN (Mask R-CNN) consists of five subnetworks: a backbone, a regional proposal network, a region CNN head, a mask head, and a mask score head. Their approach has been validated on 70 patients' images with ground truth manual contour, resulting in an efficient segmentation of breast cancer's volume from ABUS images.

X. Xie et al [16], have introduced their study for BUS image classification and segmentation using CNNs as follow: firstly, building a BUS samples' dataset (1418 normal + 1182 cancerous) labeled by three radiologists from Xiang-Ya hospital of Hunan province. then, a two-stage CAD system has been proposed for automatically breast cancer's diagnosis. X. Xie, et al have utilized a pretrained ResNet obtained by TL approach for excluding normal candidates, and then used an improved Mask R-CNN model to segment tumors accurately, resulting in efficient performance both in classification and segmentation.

M. H. Yap et al [23], have introduced their CNNs based study for automated BUS lesions detection, investigating the use of three DL approaches: Patch-based LeNet, U-Net, and FCN-AlexNet, comparing their performance against four known lesion detection algorithms (Radial Gradient Index, Multifractal Filtering, Rule-based Region Ranking, and Deformable Part Models), concluding that TL FCN-AlexNet achieved the best results.

S. Hussain et al. [24], have proposed contextual level-set method for breast tumor segmentation, by developing an encoder-decoder architecture network such as UNet to learn high-level contextual features with semantic information, then the contextual level set method has been introduced for incorporating the contextual energy term, the proposed term can embed the high-level contextual knowledge into the level set framework, then more discriminative

information been directly related to class labels (instead of original intensity) can be provided by the learned contextual features with semantic information.

W. Al-Dhabyani et al [33], have introduced their study about data augmentation and classification for breast masses in BUS images by DL approaches, validating their work by two different approaches (CNN and TL) with and without augmentation. Traditional and Generative Adversarial Network (GAN) based augmentation have been applied in their work, achieving an efficient performance resulted from integrating traditional with GAN-based augmentation.

Reena M. Roy and Ameer P.M. [34], have introduced their approach of Segmentation of leukocyte by, employing an SS technique uses DeepLabv3+ architecture with ResNet-50 as a feature extractor network, carrying out their experiments on three different public datasets consisting of five categories of white blood cells, asserting their model effectiveness by a 10-fold cross-validation, achieving an efficient segmentation performance.

L. Ahmed et al [35], have introduced their breast cancer SS study of images data practices using deep neural network, their study validated by two mammography's images datasets (Mammographic Image Analysis Society (MIAS), and Curated Breast Imaging Subset of (Digital Database for Screening Mammography) (CBIS-DDSM)), proposing a preprocessing mechanism for removing noise, artifacts and muscle region which could cause a high false positive rate.

R. Yang and Y. Yu [36], have introduced their review demonstrating most of the important roles played by artificial CNNs and their extension algorithms in SS, object detection, and medical imaging classification.

C. Iwendi et al [37], have proposed their study about the role of an Adaptive Neuro-Fuzzy Inference System (ANFIS) into Classification of Coronavirus Disease (COVID-19) individuals, introducing a system to analyze and classify the predictions produced from virus's symptoms, aiming to help in COVID-19's early detection.

S. Abbas et al [38], have introduced an approach named BCD-WERT for breast cancer detection utilizing Whale Optimization Algorithm (WOA) and extremely randomized tree for enhanced selection and classification of features. When been compared with eight different machine learning (ML) algorithms (Support Vector Machine (SVM), Random Forest, Kernel SVM, Decision Tree, Logistic Regression (LR), Stochastic Gradient Descent (SGD), Gaussian Naive Bayes (GNB) and k-Nearest Neighbor (KNN)), BCD-WERT has achieved an outperformance over all the eight.

## Materials and methods

### 1 Fuzzy Intensification Operator (FIO) based image enhancement

The first introduction to fuzzy sets has been in 1965 by L. A. Zadeh, who has defined what called a fuzzy set as a characterized objects' class with membership function often ranging between zero and one [39] Here, we applied a fuzzy FIO based method for contrast enhancement [25–30]. The applied FIO based method is consisted of three steps as follow:

**1.1 Fuzzification.** The first step is a transform of the image from the spatial domain into fuzzy domain producing a "fuzzy image" by a pixel-by-pixel fuzzification process described by (1):

$$\mu\left(i,j\right) = \frac{Input\left(i,j\right) - Min}{Max - Min} \tag{1}$$

*Where;* $\mu\left(i,j\right)$ represents the resulted fuzzy membership's value calculated for the pixel value allocated in row number (i) and a column number (j) in the input image, *Input (i, j)* represents

the input image's pixel value allocated in row number (i) and a column number (j), Min: is the minimum pixel value in the input image, and Max: is the maximum pixel value in the input image.

**1.2 Applying intensification operator.**   The second step is applying an intensifier operator according to (2) to calculate the modified membership value μ' for each pixel in the "fuzzy image" producing the modified membership fuzzy image.

$$\mu'(i,j)\begin{cases} 2*\mu\,(i,j)^2 & 0 \le \mu\,(i,j) \le 0.5 \\ 1-2(1-\mu\,(i,j)^2) & 0.5 < \mu\,(i,j) \le 1.0 \end{cases} \tag{2}$$

*Where;* μ' (i, j) represents the modified membership μ' corresponding to the pixel μ (i, j) in the fuzzy image after applying the intensification process.

**1.3 De-fuzzification.**   Finally, the third step is a transformation of the enhanced modified membership fuzzy image (produced from the last step) into spatial domain by the de-fuzzification process through the following relation (3).

$$Output(i,j) = Min + \mu'(i,j) * (Max - Min) \tag{3}$$

*Where; Output (i, j)* represents the final FIO based enhanced image's pixel value in gray scale allocated in row number (i) and a column number (j).

A Sample MATLAB program applying the described FIO procedure in 1.1, 1.2, and 1.3 can be founded into supporting information section (**S2 Data**).

## 2 The data set used

A collection of 400 BUS with tumor images and their 400 ground truth images taken from [31] has been utilized in our study. W. Al-Dhabyani, et al, have collected all the dataset in [31] for BUS images from a variety of women in ages between 25 and 75 years old in 2018, by LOGIQ E9 ultrasound and LOGIQ E9 Agile ultrasound system at Baheya Hospital for Early Detection & Treatment of Women's Cancer, Cairo, Egypt. The 800 images taken from [31] has been utilized as follow:

■ 200 images for BUS with a benign breast cancer and their 200 ground truth images.

■ 200 images for BUS with a malignant breast cancer and their 200 ground truth images.

All of the 400 BUS images have been resized to 128 by 128 by 3 to be appropriate for the input layer for the eight CNN semantic segmentation networks utilized in our study.

All of the 400 ground truth images have been processed so that the black background (represents normal tissue) has a value of one "1" and the tumor's region has a value of two "2", on a grayscale (from 0 to 255), and resized to 128 by 128 to be appropriate to be compared by the output of the applied semantic segmentation process. The 400 BUS with tumor images has been enhanced by an FIO based method for contrast enhancement (as been demonstrated in the previous subsections), producing another 400 enhanced images with size 128 by 128 by 3.

Then we have a new dataset contains 1200 images divided as:

1. 200 BUS images (size: 128 by 128 by 3) with a benign cancer (Original_Benign).

2. 200 Fuzzy-enhanced BUS' images (size: 128 by 128 by 3) with a benign cancer (Fuzzy_Benign).

3. 200 ground truth images (size: 128 by 128) for benign cancer (Ground_Truth_Benign).

4. 200 BUS images (size: 128 by 128 by 3) with a malignant cancer (Original_Malignant).

5. 200 Fuzzy-enhanced BUS' images (size: 128 by 128 by 3) with a malignant cancer (Fuzzy_Malignant).

6. 200 ground truth images (size: 128 by 128) for malignant cancer (Ground_Truth_Malignant).

All the 1200 images are arranged each different 200 images in one folder. The first 600 images (Original_Benign, Fuzzy_Benign, and Ground_Truth_Benign) are labeled for the same 200 benign BUS images and saved in one folder (Benign). The last 600 images (Original_Malignant, Fuzzy_ Malignant, and Ground_Truth_ Malignant) are labeled for the same 200 malignant BUS images and saved in one folder (Malignant). The final two folders (Benign and Malignant) are combined in one folder called **MT_small_dataset.** This dataset folder can be founded in the supporting information section (S1 Data) and also is available for all researchers at: https://www.kaggle.com/mohammedtgadallah/mt-small-dataset.

## 3 Automatic SS

CNNs, recently, have achieved a noticeable success in automatic SS of tumors in BUS images [11–13, 17, 18, 23, 40]. Eight (in this paper it is referred to eight by: **X 8**) well-known CNN-based SS models taken from [11] have been utilized in our study: FCN with AlexNet network, UNet network, SegNet using VGG16, SegNet using VGG19, DeepLabV3+ using ResNet18, DeepLabV3+ using ResNet50, DeepLabV3+ using MobileNet-V2, and DeepLabV3+ using Xception networks. The Eight CNN-based SS schemes has been utilized for comparing its quantitative and qualitative performance in automatic semantic segmentation before and after applying the demonstrated FIO based enhancement scheme. SS process has been applied through the eight CNNs in two modes (S3 Data):

**3.1 Batch images' processing mode.** In batch mode the input to the segmentation network has been divided into four datastores (batches) for:

One datastore for 200 benign BUS images.

One for 200 malignant BUS images.

One for 200 benign after being enhanced by fuzzy preprocessing.

One for 200 malignant after being enhanced by fuzzy preprocessing.

Each batch has been segmented as a one datastore by MATLAB's inherent function "semanticseg" [32].

**3.2 One by one image's processing mode.** In one by one mode the input to the segmentation network is only one image at a time. So, to segment 200 images; the MATLAB's inherent function "semanticseg" [32] has been repeated 200 times. This method is more accurate than the batch mode.

## 4 Performance evaluation

In order to evaluate the performance of the segmentation process, several metrics can be used [19–21, 34, 41–43]. This paper has utilized three measures: GA, Mean IoU (Jaccard Index), and Mean BF (Boundary F1) Score. "Mean" means the average of the metric of all classes in all images. There are two classes "Tumor" and "Normal Tissue". The two classes represented by gray level pixel values from "0" to "255" as: "2" for "Tumor" and "1" for "Normal Tissue". Calculations of the three metrics are done by inherent function on MATLAB

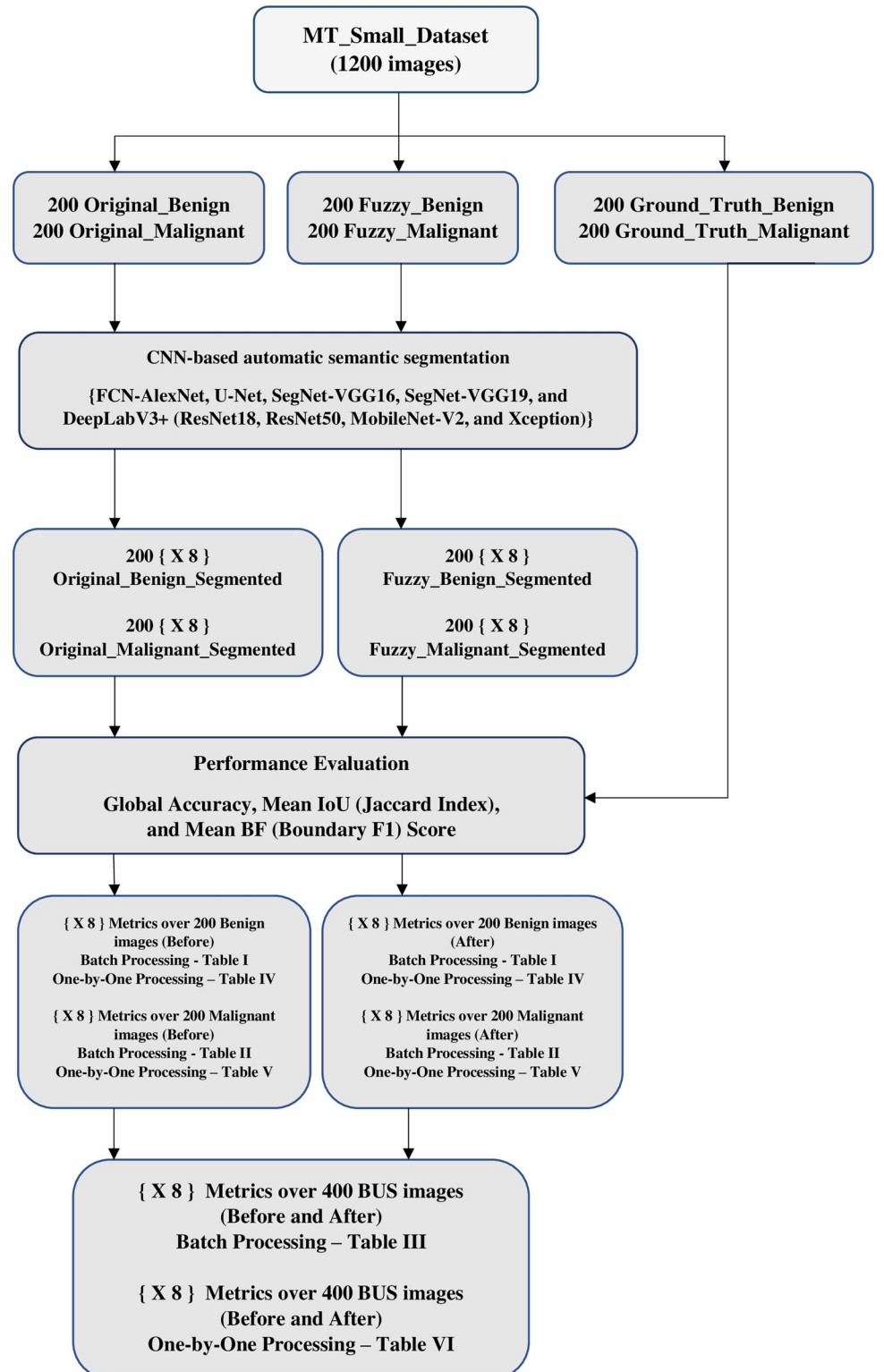

**Fig 1. Illustrative Flowchart for the proposed framework.**

**Table 1. Semantic segmentation evaluation metrics for 200 benign BUS images before and after applying fuzzy enhancement (based on batch images' processing).**

| Semantic Segmentation Network | Global Accuracy | | Mean IoU (Jaccard Index) | | Mean BF (Boundary F1) Score | |
|---|---|---|---|---|---|---|
| | % Before | % After | % Before | % After | % Before | % After |
| FCN - AlexNet | 90.62 | 97.22 | 49.11 | 78.60 | 44.68 | 63.78 |
| U-Net | 91.08 | 97.40 | 49.09 | 78.65 | 44.93 | 77.22 |
| SegNet - VGG16 | 90.99 | 98.14 | 49.13 | 83.77 | 45.31 | 82.31 |
| SegNet - VGG19 | 90.88 | 98.16 | 49.13 | 84.00 | 45.26 | 82.66 |
| DeepLabV3+ - ResNet18 | 90.54 | 98.01 | 48.87 | 83.84 | 45.14 | 81.91 |
| DeepLabV3+ - ResNet50 | 90.47 | 98.38 | 48.80 | 85.70 | 45.10 | 83.49 |
| DeepLabV3+ - MobileNet-V2 | 90.94 | 97.78 | 48.99 | 81.27 | 45.13 | 80.66 |
| DeepLabV3+ - Xception | 90.81 | 98.11 | 49.08 | 83.99 | 45.10 | 82.01 |

"evluateSemanticSegmentation" [44]. Where;

Global Accuracy (GA)

$$= \frac{\text{Number of True classified pixels (regardless of class)}}{\text{Total Number of Pixels}} = \frac{TP + TN}{TP + TN + FP + FN} \quad (4)$$

$$\text{IoU (Jaccard Index)} = \frac{\text{Intersection(I)}}{\text{Union(U)}} = \frac{TP}{TP + FP + FN} \quad (5)$$

$$\text{BF (Boundary F1)Score} = \frac{2 * \text{Precision} * \text{Recall}}{\text{Precision} + \text{Recall}} = \frac{2 * TP}{2 * TP + FP + FN} \quad (6)$$

Where;

$$\text{Precision} = \frac{TP}{TP + FP}, \ \text{Recall (Sensitivity)} = \frac{TP}{TP + FN},$$

*TP*: *True Positive*, *TN*: *True Negative*, *FP*: *False Positive*, *FN*: *False Negative*

The three-performance metrics defined in (4), (5), and (6) were measured in between overall 200 sample segmented images' datastore (the output from the automatic segmentation process) and their equivalent 200 ground truth images' datastore (reference) as illustrated below:

1. 200 Original_Benign_Segmented (Before) *vs* 200 Ground_Truth_Benign.

2. 200 Fuzzy_Benign_ Segmented (After) *vs* 200 Ground_Truth_Benign.

**Table 2. Semantic segmentation evaluation metrics for 200 malignant BUS images before and after applying fuzzy enhancement (based on batch images' processing).**

| Semantic Segmentation Network | Global Accuracy | | Mean IoU (Jaccard Index) | | Mean BF (Boundary F1) Score | |
|---|---|---|---|---|---|---|
| | % Before | % After | % Before | % After | % Before | % After |
| FCN - AlexNet | 81.34 | 92.06 | 50.08 | 72.21 | 40.43 | 50.89 |
| U-Net | 83.09 | 91.74 | 50.58 | 69.39 | 40.91 | 54.50 |
| SegNet - VGG16 | 81.65 | 93.53 | 50.45 | 76.91 | 39.53 | 57.42 |
| SegNet - VGG19 | 81.94 | 93.53 | 50.72 | 76.57 | 40.12 | 58.80 |
| DeepLabV3+ - ResNet18 | 80.83 | 93.34 | 49.97 | 77.22 | 40.18 | 59.33 |
| DeepLabV3+ - ResNet50 | 80.07 | 93.91 | 49.34 | 77.78 | 40.07 | 60.21 |
| DeepLabV3+ - MobileNet-V2 | 81.71 | 92.80 | 50.31 | 73.81 | 40.47 | 55.93 |
| DeepLabV3+ - Xception | 80.34 | 93.02 | 50.03 | 75.39 | 39.63 | 58.06 |

**Table 3. Average quantitative evaluation metrics (based on batch images' processing) over for 400 BUS images over 8 CNN based SS models (average Tables 1 and 2).**

| Semantic Segmentation Network | Global Accuracy | | Mean IoU (Jaccard Index) | | Mean BF (Boundary F1) Score | |
|---|---|---|---|---|---|---|
| | % Before | % After | % Before | % After | % Before | %After |
| FCN - AlexNet | 85.98 | 94.64 | 49.60 | 75.41 | 42.56 | 57.34 |
| U-Net | 87.09 | 94.57 | 49.84 | 74.02 | 42.92 | 65.86 |
| SegNet - VGG16 | 86.32 | 95.84 | 49.79 | 80.34 | 42.42 | 69.87 |
| SegNet - VGG19 | 86.41 | 95.85 | 49.93 | 80.29 | 42.69 | 70.73 |
| DeepLabV3+ - ResNet18 | 85.69 | 95.68 | 49.42 | 80.53 | 42.66 | 70.62 |
| DeepLabV3+ - ResNet50 | 85.27 | 96.15 | 49.07 | 81.74 | 42.59 | 71.85 |
| DeepLabV3+ - MobileNet-V2 | 86.33 | 95.29 | 49.65 | 77.54 | 42.80 | 68.30 |
| DeepLabV3+ - Xception | 85.58 | 95.57 | 49.56 | 79.69 | 42.37 | 70.04 |
| **Average** | **86.08** | **95.45** | **49.61** | **78.70** | **42.63** | **68.08** |

3. 200 Original_Malignant_ Segmented (Before) *vs* 200 Ground_Truth_ Malignant.

4. 200 Fuzzy_ Malignant_ Segmented (After) *vs* 200 Ground_Truth_ Malignant.

Where; "Before" means that the segmentation process has been done on a 200 sample from the dataset before applying the fuzzy enhancement (Original_Benign, and Original_Malignant). Where; "After" means that the segmentation process has been done on a 200 sample from the dataset after applying the fuzzy enhancement (Fuzzy_Benign, and Fuzzy_Malignant). The four steps illustrated above have been applied eight times {X 8} each with different CNN-based SS model with two modes for images processing: batch and one by one as illustrated into subsections 3.1 and 3.2 in materials and methods' section, respectively.

## 5 The applied work on MT_small_dataset

The proposed framework is illustrated in Fig 1.

## Results and discussion

The output evaluation metrics of the batch mode for all SS operation on both benign and malignant BUS images by eight models (we have referred to 8 models by: {**x 8**}) before and after applying fuzzy enhancement to the input images' set are illustrated in Tables 1 and 2, respectively. The average results for Tables 1 and 2 are displayed in Table 3. The output evaluation metrics of the one by one image mode for all SS operation on both benign and malignant

**Table 4. Semantic segmentation evaluation metrics for 200 benign BUS images before and after applying fuzzy enhancement (based on one by one image processing).**

| Semantic Segmentation Network | Global Accuracy | | Mean IoU (Jaccard Index) | | Mean BF (Boundary F1) Score | |
|---|---|---|---|---|---|---|
| | % Before | % After | % Before | % After | % Before | % After |
| FCN - AlexNet | 98.08 | 97.22 | 84.03 | 78.60 | 68.08 | 63.78 |
| U-Net | 98.03 | 97.40 | 82.92 | 78.65 | 83.15 | 77.22 |
| SegNet - VGG16 | 98.50 | 98.14 | 86.60 | 83.75 | 85.57 | 82.30 |
| SegNet - VGG19 | 98.66 | 98.16 | 88.06 | 83.97 | 86.19 | 82.66 |
| DeepLabV3+ - ResNet18 | 98.76 | 98.01 | 89.13 | 83.84 | 86.40 | 81.91 |
| DeepLabV3+ - ResNet50 | 98.76 | 98.38 | 89.24 | 85.70 | 86.33 | 83.49 |
| DeepLabV3+ - MobileNet-V2 | 98.57 | 97.78 | 87.24 | 81.27 | 86.15 | 80.66 |
| DeepLabV3+ - Xception | 98.51 | 98.11 | 86.98 | 83.99 | 85.75 | 82.01 |

**Table 5. Semantic segmentation evaluation metrics for 200 malignant BUS images before and after applying fuzzy enhancement (based on one by one image processing).**

| Semantic Segmentation Network | Global Accuracy | | Mean IoU (Jaccard Index) | | Mean BF (Boundary F1) Score | |
|---|---|---|---|---|---|---|
| | % Before | % After | % Before | % After | % Before | % After |
| FCN - AlexNet | 92.77 | 92.06 | 75.34 | 72.21 | 51.83 | 50.89 |
| U-Net | 92.64 | 91.74 | 73.03 | 69.39 | 56.29 | 54.50 |
| SegNet - VGG16 | 94.54 | 93.51 | 80.40 | 76.86 | 59.67 | 57.41 |
| SegNet - VGG19 | 94.68 | 93.52 | 80.69 | 76.56 | 60.49 | 58.83 |
| DeepLabV3+ - ResNet18 | 94.82 | 93.34 | 81.83 | 77.22 | 61.88 | 59.33 |
| DeepLabV3+ - ResNet50 | 94.62 | 93.91 | 81.58 | 77.78 | 60.03 | 60.21 |
| DeepLabV3+ - MobileNet-V2 | 93.82 | 92.80 | 78.09 | 73.81 | 60.17 | 55.93 |
| DeepLabV3+ - Xception | 93.25 | 93.02 | 77.80 | 75.39 | 59.93 | 58.05 |

BUS images by eight models {**x 8**} before and after applying fuzzy enhancement to the input images' set are illustrated in Tables 4 and 5, respectively. The average results for Tables 4 and 5 are displayed in Table 6.

Looking at Tables 1 and 2 for batch processing, it can be noticed that: the global accuracy, mean IoU, and mean F1 score results from the automatic segmentation's performance evaluation, are all increased obviously after applying the preprocessing fuzzy enhancement step. This obvious quantitative enhancement assures the success of our proposed approach in batch automatic semantic segmentation. For making an overall view of the quantitative metrics over a 400 BUS images on eight CNN based SS models, Tables 1 and 2 are merged and averaged producing a concentrated view for the quantitative batch frame work represented in Table 3.

Figs 2–4 represent three illustrative charts for the average results for; global accuracy, mean IoU, and mean F1 score, in percent for the eight CNNs based SS approaches over 400 BUS images before and after applying the proposed batch processing approach. Figs 2–4 show clearly the enhancement developed for global accuracy, mean IoU, and mean F1 score, after applying the proposed scheme in this paper for batch segmentation.

Qualitative results for the batch segmentation process based on applying the preprocessing fuzzy enhancement shown a noticeable enhancement in the segmentation of breast tumors in BUS images. A sample from the qualitative visual batch segmentation's results is illustrated in Figs 5–8. Fig 5 represents eight samples from 200 BUS benign images' results for the batch automatic SS applied using DeepLabV3+ / ResNet18 and shows their original base image before applying segmentation and also shows their ground truth images for qualitative visual

**Table 6. Average quantitative evaluation metrics (based on one by one image processing) over for 400 BUS images over 8 CNN based SS models (average Tables 4 and 5).**

| Semantic Segmentation Network | Global Accuracy | | Mean IoU (Jaccard Index) | | Mean BF (Boundary F1) Score | |
|---|---|---|---|---|---|---|
| | % Before | % After | % Before | % After | % Before | % After |
| FCN - AlexNet | 95.43 | 94.64 | 79.69 | 75.41 | 59.96 | 57.34 |
| U-Net | 95.34 | 94.57 | 77.98 | 74.02 | 69.72 | 65.86 |
| SegNet - VGG16 | 96.52 | 95.83 | 83.50 | 80.31 | 72.62 | 69.86 |
| SegNet - VGG19 | 96.67 | 95.84 | 84.38 | 80.27 | 73.34 | 70.75 |
| DeepLabV3+ - ResNet18 | 96.79 | 95.68 | 85.48 | 80.53 | 74.14 | 70.62 |
| DeepLabV3+ - ResNet50 | 96.69 | 96.15 | 85.41 | 81.74 | 73.18 | 71.85 |
| DeepLabV3+ - MobileNet-V2 | 96.20 | 95.29 | 82.67 | 77.54 | 73.16 | 68.30 |
| DeepLabV3+ - Xception | 95.88 | 95.57 | 82.39 | 79.69 | 72.84 | 70.03 |
| **Average** | **96.19** | **95.44** | **82.69** | **78.69** | **71.12** | **68.07** |

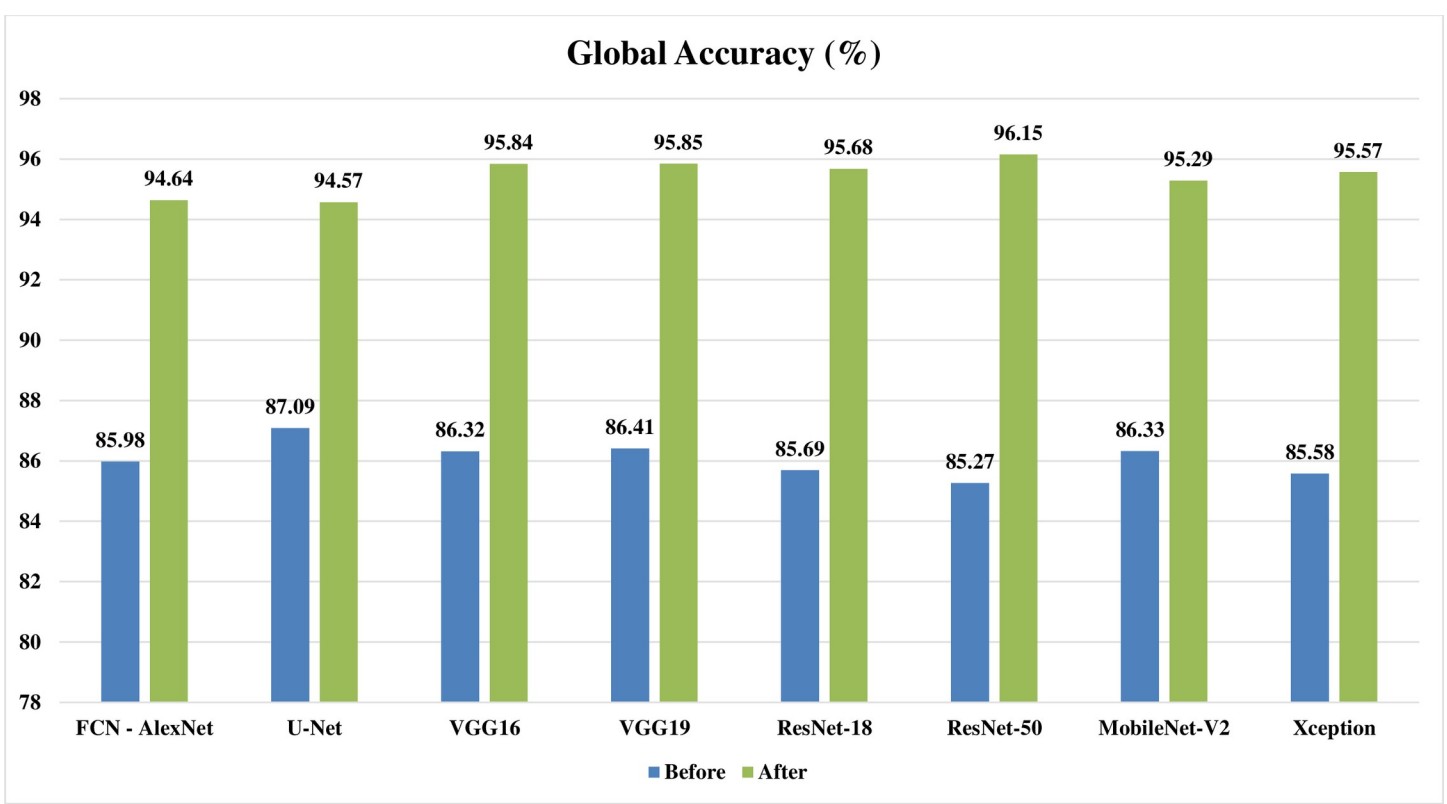

**Fig 2. Illustrative chart for the average global accuracy results in percent for eight CNNs based SS over 400 BUS images presented in Table 3 (batch processing).**

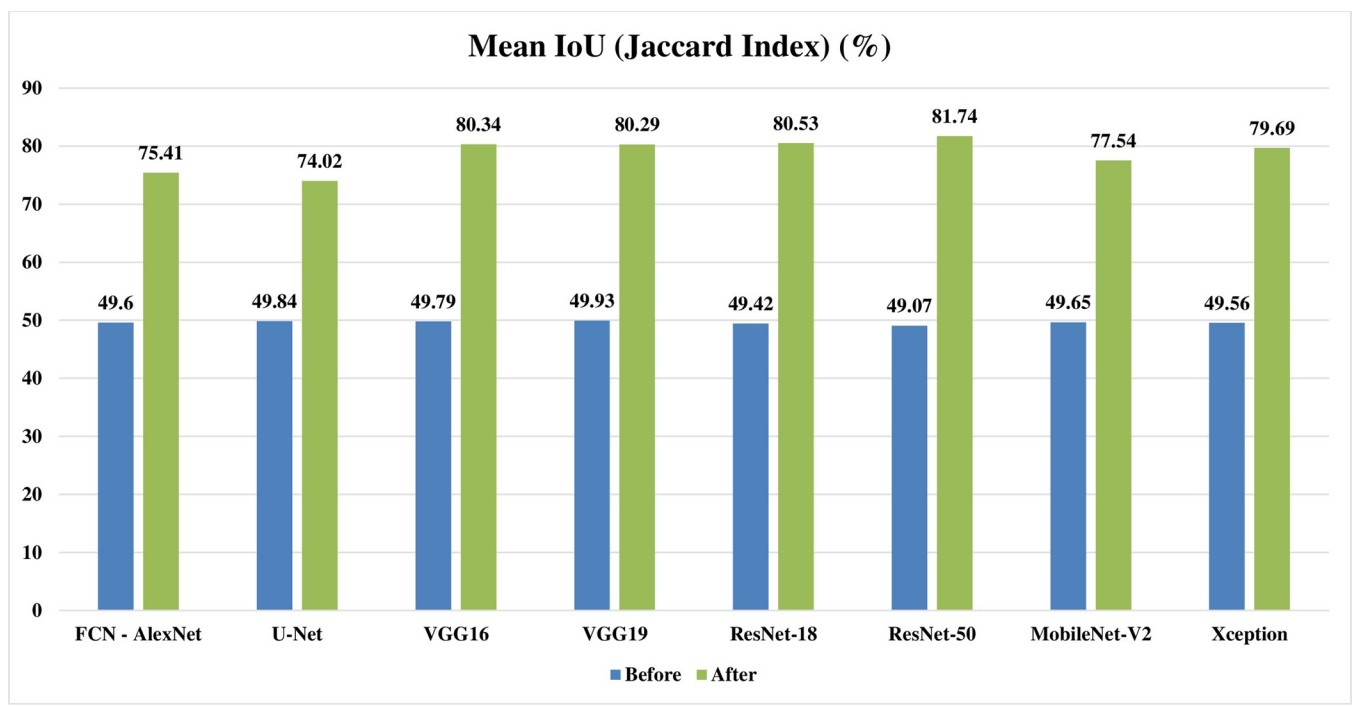

**Fig 3. Illustrative chart for the average mean IoU (Jaccard Index) results in percent for eight CNNs based SS over 400 BUS images presented in Table 3 (batch processing).**

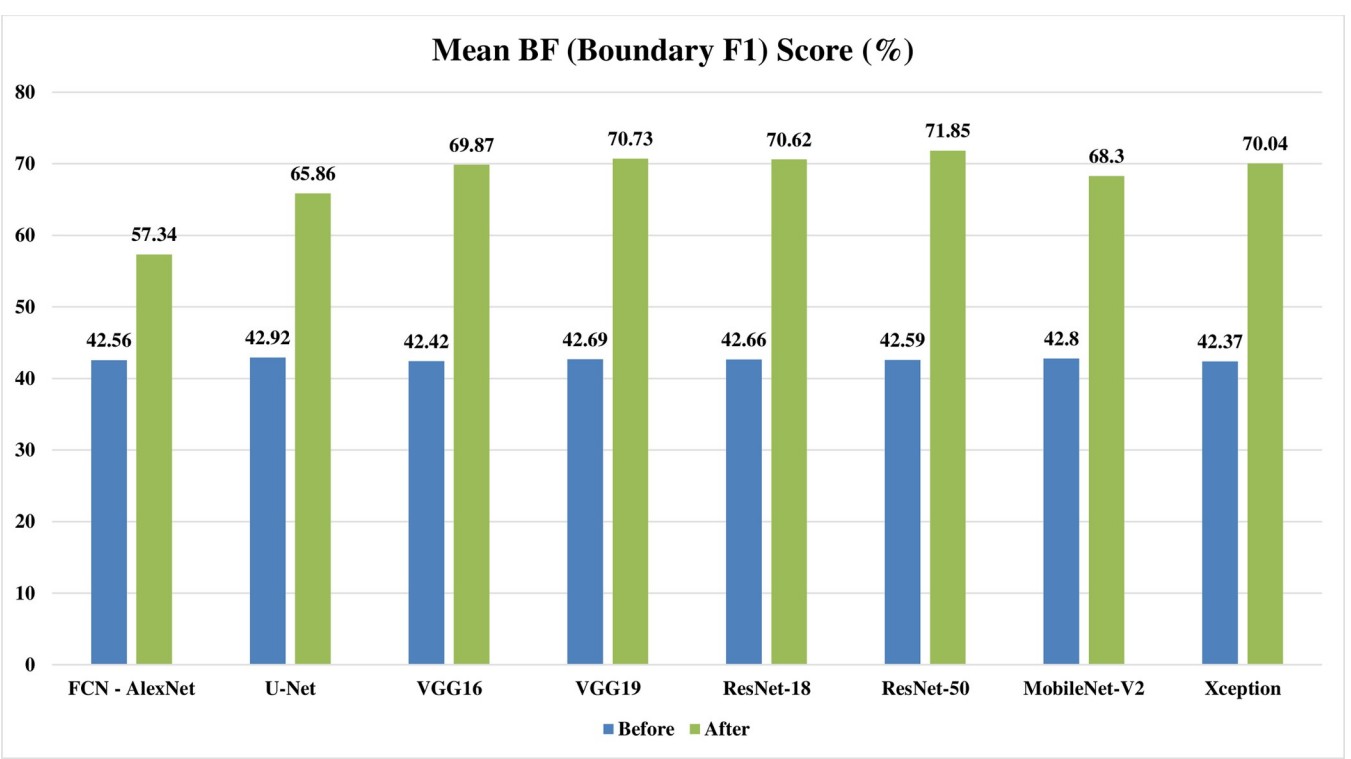

**Fig 4. Illustrative chart for the average mean BF (Boundary F1) Score results in percent for eight CNNs based SS over 400 BUS images presented in Table 3 (batch processing).**

comparing. Each sample consist of four images: 1st left image is the original image from the prepared small dataset, 2nd image is the segmentation results without applying fuzzy enhancement, 3rd image is the segmentation result after applying fuzzy based enhancement to the input image, and finally 4th image is the original ground truth founded here for a fair visual qualitative assessment in-between it and the prior two segmented images (2nd image before and 3rd image after applying fuzzy preprocessing enhancement). Fig 6 demonstrates eight

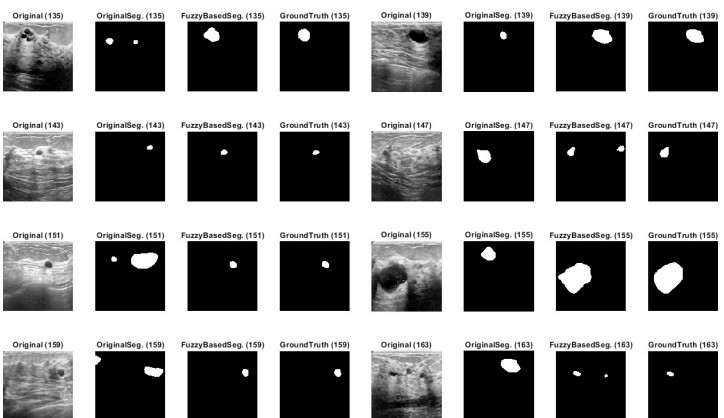

**Fig 5. Eight samples from 200 BUS benign images' results for ResNet18 (batch processing).** Each sample consist of four images: 1st the original image from the prepared small dataset, 2nd the segmentation results without applying fuzzy enhancement, 3rd the segmentation result after applying fuzzy based enhancement to the input image, and 4th the original ground truth.

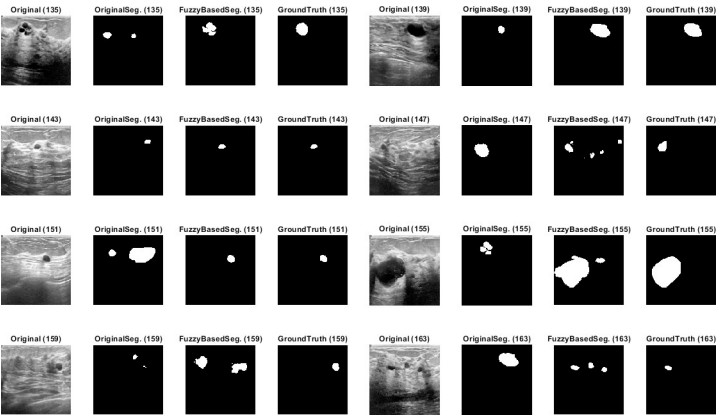

**Fig 6. Eight samples from 200 BUS benign images' results for U-Net (batch processing).** Each sample consist of four images: 1st the original image from the prepared small dataset, 2nd the segmentation results without applying fuzzy enhancement, 3rd the segmentation result after applying fuzzy based enhancement to the input image, and 4th the original ground truth.

samples from 200 BUS benign images' segmentation results applied by U-Net for the same eight images shown in Fig 2 by ResNet18. Fig 7 represents eight samples from 200 BUS malignant images' results obtained by ResNet18 based SS before and after applying fuzzy preprocessing, the base eight images before segmentation, and the ground truth reference for visual comparing. Fig 8 represents eight samples from 200 BUS malignant images' results obtained by U-Net based automatic SS. Fig 8 displays the applied approach on the same eight base images founded in Fig 7 but with applying U-Net instead of ResNet18. The overall quantitative and qualitative results, for batch processing, assured the success of the proposed approach into enhancing BUS images' automatic SS. Hence, better diagnosis.

Looking at Tables 4 and 5 for one by one processing, it can be noticed that: the global accuracy, mean IoU, and mean F1 score results from the automatic segmentation's performance evaluation, are all decreased obviously after applying the preprocessing fuzzy enhancement step. This obvious quantitative diminishment assures the failure of our proposed approach in one by one automatic SS mode. Looking at Figs 9–11, three illustrative charts for the average

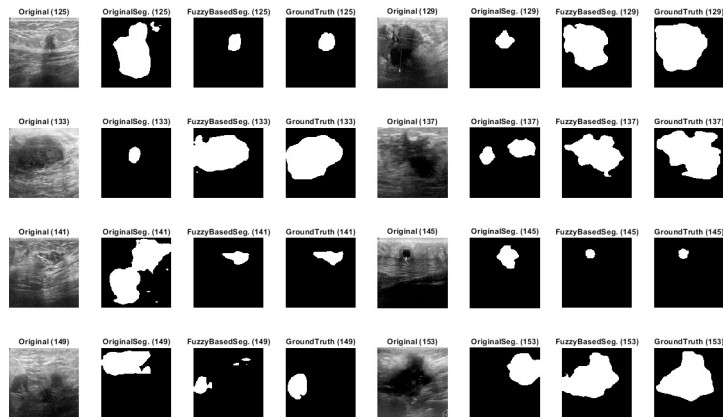

**Fig 7. Eight samples from 200 BUS malignant images' results for ResNet18 (batch processing).** Each sample consist of four images: 1st the original image from the prepared small dataset, 2nd the segmentation results without applying fuzzy enhancement, 3rd the segmentation result after applying fuzzy based enhancement to the input image, and 4th the original ground truth.

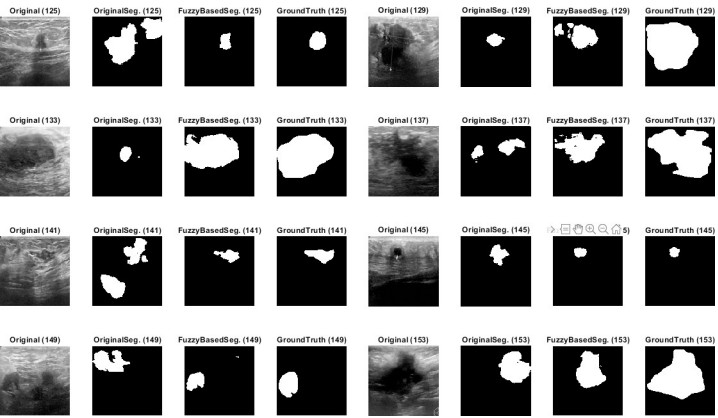

**Fig 8. Eight samples from 200 BUS malignant images' results for U-Net (batch processing).** Each sample consist of four images: 1st the original image from the prepared small dataset, 2nd the segmentation results without applying fuzzy enhancement, 3rd the segmentation result after applying fuzzy based enhancement to the input image, and 4th the original ground truth.

results for one by one processing mode are illustrated for: GA, mean IoU, and mean F1 score, taken from Table 5, in percent for the eight CNNs based SS over 400 BUS images before and after applying the proposed batch processing approach. Figs 9–11 show clearly the diminishment induced for GA, mean IoU, and mean F1 score, after applying the proposed fuzzy based scheme. The diminishment is clearly illustrated qualitatively in the samples shown in Figs 12–

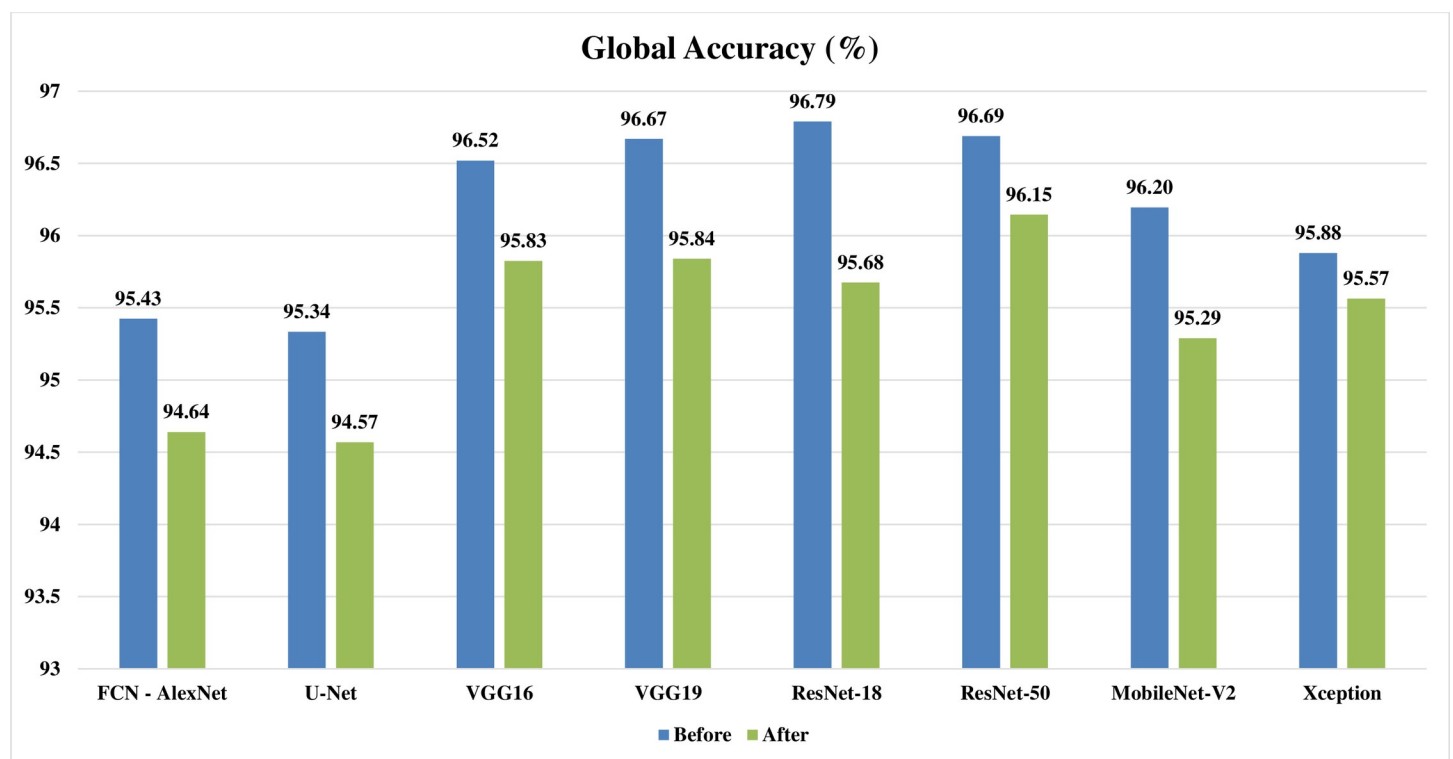

**Fig 9. Illustrative chart for the average global accuracy results in percent for eight CNNs based SS over 400 BUS images presented in Table 6 (one by one processing).**

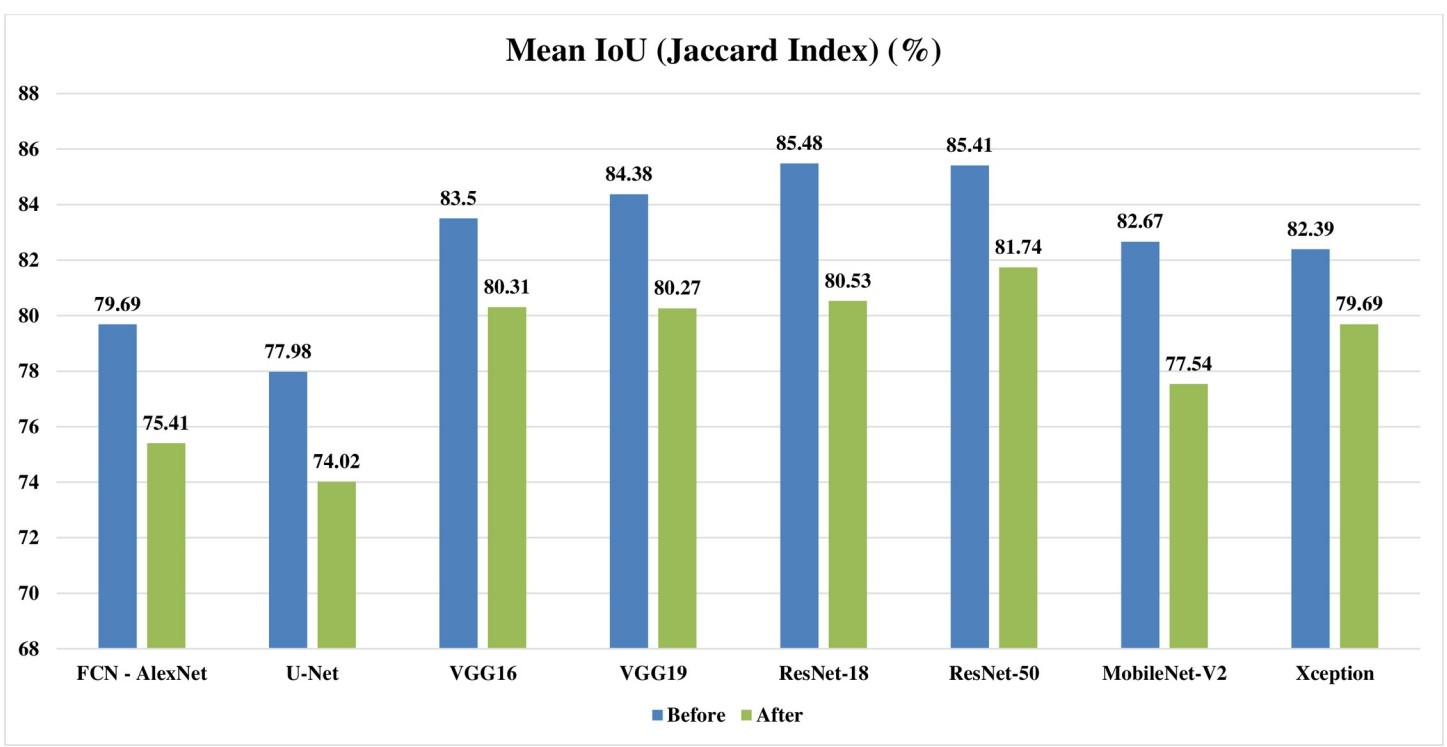

**Fig 10. Illustrative chart for the average mean IoU (Jaccard Index) results in percent for eight CNNs based SS over 400 BUS images presented in Table 6 (one by one processing).**

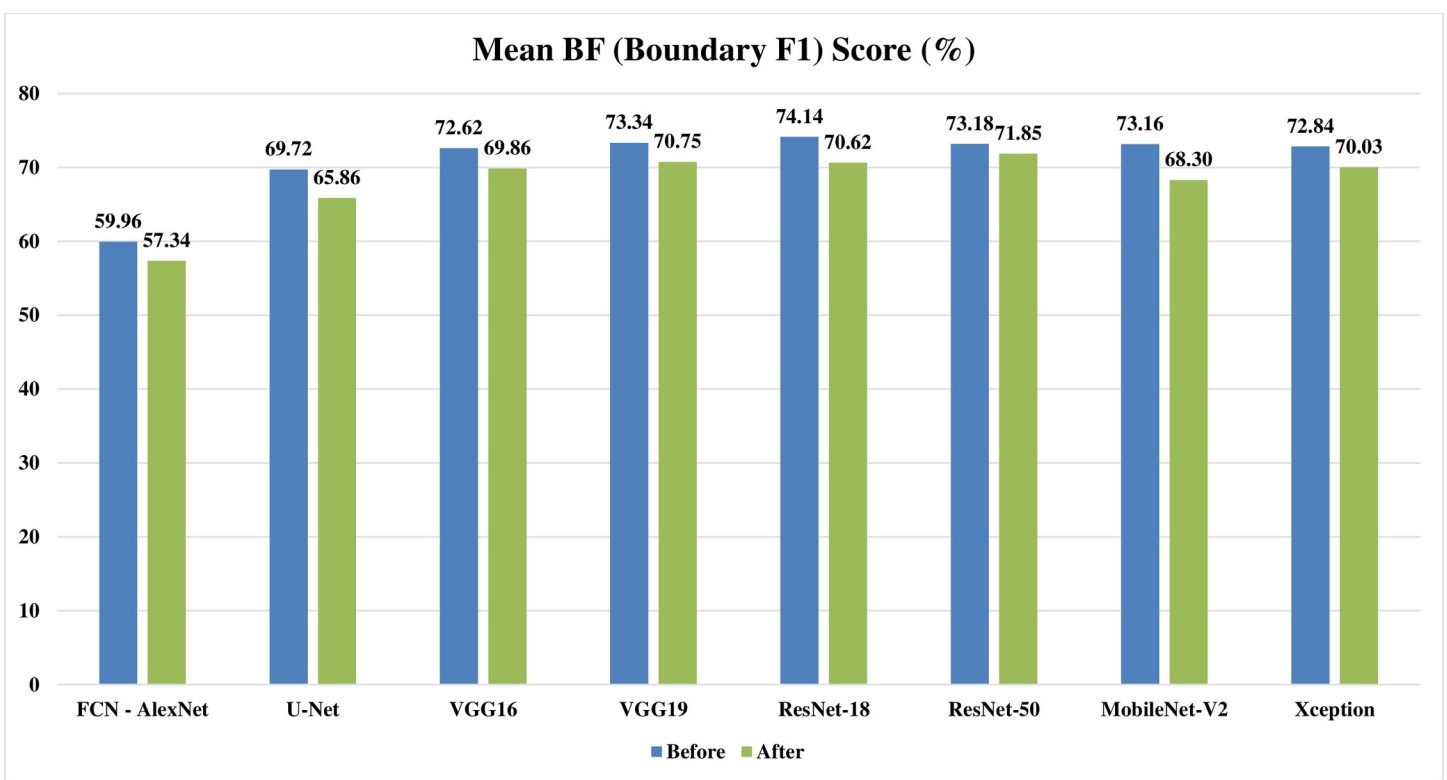

**Fig 11. Illustrative chart for the average mean BF (Boundary F1) Score results in percent for eight CNNs based SS over 400 BUS images presented in Table 6 (one by one processing).**

## Sample - Benign

**Fig 12. Eight samples from 200 BUS benign images' results for ResNet18 (one by one processing).** Each sample consist of four images: 1st the original image from the prepared small dataset, 2nd the segmentation results without applying fuzzy enhancement, 3rd the segmentation result after applying fuzzy based enhancement to the input image, and 4th the original ground truth.

15 which demonstrate a more disrupted performance after applying fuzzy preprocessing when compared to the original segmented images by only CNN based SS. **So, for one by one image processing mode, where there is no need for batch processing, it is recommended not to use the proposed fuzzy based SS scheme.**

## Conclusion

Efficient automatic characterization of tumors in BUS batch images has been proposed by combining a preprocessing fuzzy enhancement step before starting a known CNN based SS

## Sample - Benign

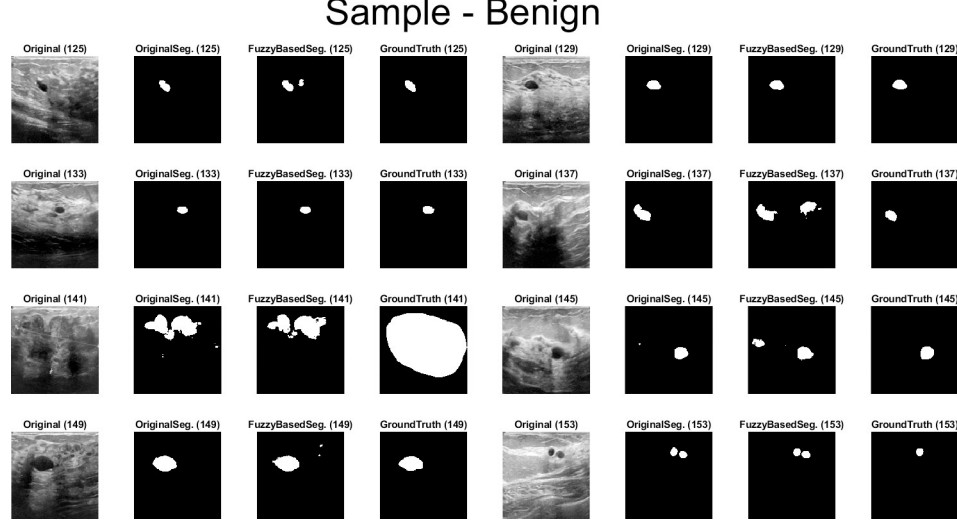

**Fig 13. Eight samples from 200 BUS benign images' results for U-Net (one by one processing).** Each sample consist of four images: 1st the original image from the prepared small dataset, 2nd the segmentation results without applying fuzzy enhancement, 3rd the segmentation result after applying fuzzy based enhancement to the input image, and 4th the original ground truth.

## Sample - Malignant

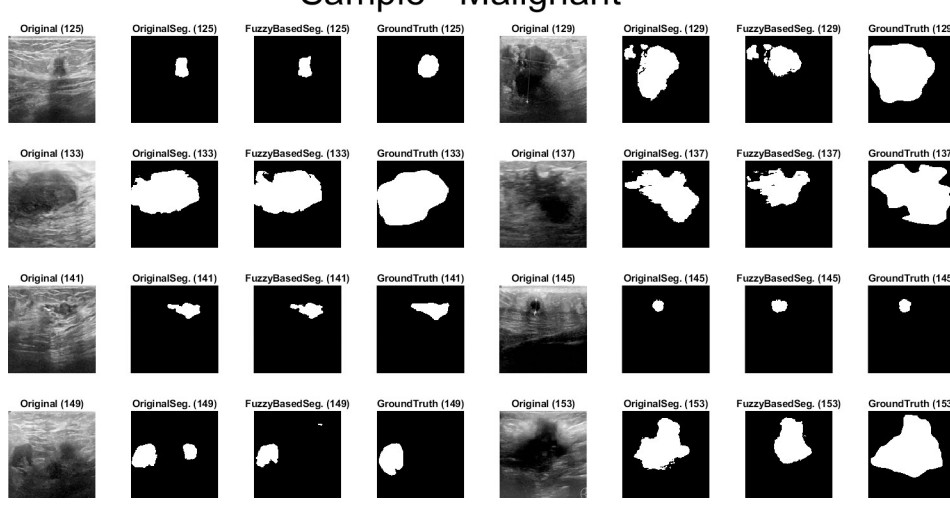

**Fig 14. Eight samples from 200 BUS malignant images' results for ResNet18 (one by one processing).** Each sample consist of four images: 1st the original image from the prepared small dataset, 2nd the segmentation results without applying fuzzy enhancement, 3rd the segmentation result after applying fuzzy based enhancement to the input image, and 4th the original ground truth.

model. Preprocessing enhancement step based on fuzzy intensification operator has been applied to increase the contrast of the BUS batch images and reduce the fuzziness of them. Eight CNN based SS models have been utilized: FCN-AlexNet, UNet, SegNet-VGG16, SegNet-VGG19, and DeepLabV3+(ResNet18, ResNet50, MobileNet-V2, and Xception. The study validated on a modified small dataset (MT_Small_Dataset) containing 1200 images: 400 cancerous BUS (128 by 128 by 3), the same 400 after applying fuzzy based contrast enhancement (128 by 128 by 3), and 400 image ground truth (128 by 128). The study has been applied in two

## Sample - Malignant

**Fig 15. Eight samples from 200 BUS malignant images' results for U-Net (one by one processing).** Each sample consist of four images: 1st the original image from the prepared small dataset, 2nd the segmentation results without applying fuzzy enhancement, 3rd the segmentation result after applying fuzzy based enhancement to the input image, and 4th the original ground truth.

different modes: batch processing and one by one image processing mode. Three known metrics have been utilized for quantitative evaluation of the proposed approach: mean IoU, mean BF, and GA. Quantitative and Qualitative performance assessment assured the success of the proposed approach to achieve an efficient automatic SS of tumors in BUS batch images. Experimental results for batch processing mode prove that; our proposed method achieved better performance in discerning specific ROI in comparing to different well-known CNN based SS models without FL based pre-processing step. Generalization of the proposed approach for batch image processing could be applied as an effective automatic SS approach for all images including biomedical imaging scans. While, utilizing the proposed approach in one by one image processing mode has no development neither quantitatively nor qualitatively. Moreover, in one by one image processing mode the proposed scheme has disrupted the SS process's efficiency. So, it is recommended not to use the proposed scheme in a one by one image mode. The proposed scheme may be useful only when a batch processing mode is needed. MT_Small_Dataset (S1 Data) is available for all researchers at: https://www.kaggle.com/mohammedtgadallah/mt-small-dataset

## Supporting information

**S1 Data. Breast cancer's ultrasound images dataset (segmentation and classification).**
https://www.kaggle.com/mohammedtgadallah/mt-small-dataset.
(RAR)

**S2 Data. A sample FIO program by MATLAB.**
(RAR)

**S3 Data. An explanation code by MATLAB for the two processing modes mentioned in the paper (one by one–batch).**
(M)

## Author Contributions

**Conceptualization:** Samir M. Badawy, Alaa A. Hefnawy, Mohammed T. GadAllah, Ghada M. El-Banby.

**Data curation:** Alaa A. Hefnawy, Mohammed T. GadAllah.

**Formal analysis:** Mohammed T. GadAllah, Ghada M. El-Banby.

**Investigation:** Mohammed T. GadAllah.

**Methodology:** Alaa A. Hefnawy, Mohammed T. GadAllah, Ghada M. El-Banby.

**Resources:** Hassan E. Zidan, Mohammed T. GadAllah, Ghada M. El-Banby.

**Software:** Mohammed T. GadAllah.

**Supervision:** Samir M. Badawy, Abd El-Naser A. Mohamed, Alaa A. Hefnawy, Hassan E. Zidan, Ghada M. El-Banby.

**Validation:** Alaa A. Hefnawy, Mohammed T. GadAllah.

**Visualization:** Alaa A. Hefnawy, Mohammed T. GadAllah, Ghada M. El-Banby.

**Writing – original draft:** Alaa A. Hefnawy, Mohammed T. GadAllah, Ghada M. El-Banby.

**Writing – review & editing:** Alaa A. Hefnawy, Mohammed T. GadAllah, Ghada M. El-Banby.

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
