## [Decision Letter · Decision Letter 0]

19 Apr 2021

PONE-D-21-09994

Automatic Semantic Segmentation of Breast Tumors in Ultrasound Images Based on Combining Fuzzy Logic and Deep Learning – A Feasibility Study

PLOS ONE

Dear Dr. GadAllah,

Thank you for submitting your manuscript to PLOS ONE. After careful consideration, we feel that it has merit but does not fully meet PLOS ONE’s publication criteria as it currently stands. Therefore, we invite you to submit a revised version of the manuscript that addresses the points raised during the review process.

Based on the suggestions from the reviewers and my own observation I recommend minor revisions for this paper.

We look forward to receiving your revised manuscript.

Kind regards,

Thippa Reddy Gadekallu

Academic Editor

PLOS ONE

Journal Requirements:

We suggest you thoroughly copyedit your manuscript for language usage, spelling, and grammar. If you do not know anyone who can help you do this, you may wish to consider employing a professional scientific editing service. 

We noted in your submission details that a portion of your manuscript may have been presented or published elsewhere. „Our submitted manuscript intitled: “Automatic Semantic Segmentation of Breast Tumors in Ultrasound Images Based on Combining Fuzzy Logic and Deep Learning – A Feasibility Study”, is a replacement to the submission No. PONE-D-21-07670 ( Efficient Automatic Semantic Segmentation of Breast Tumors in Ultrasound Images Based on Combining Fuzzy Logic and Deep Learning  ). „ Please clarify whether this publication was peer-reviewed and formally published. If this work was previously peer-reviewed and published, in the cover letter please provide the reason that this work does not constitute dual publication and should be included in the current manuscript.

Reviewers' comments:

Reviewer's Responses to Questions

**Comments to the Author**

1. Is the manuscript technically sound, and do the data support the conclusions?

Reviewer #1: Yes

Reviewer #2: Yes

2. Has the statistical analysis been performed appropriately and rigorously? 

Reviewer #1: Yes

Reviewer #2: Yes

3. Have the authors made all data underlying the findings in their manuscript fully available?

Reviewer #1: Yes

Reviewer #2: Yes

4. Is the manuscript presented in an intelligible fashion and written in standard English?

Reviewer #1: Yes

Reviewer #2: Yes

5. Review Comments to the Author

Reviewer #1: According to my understanding, this paper presentse a combination of fuzzy logic and deep learning for automatic semantic segmentation (SS) of tumors in breast ultrasound (BUS) images is proposed.

- The abbrivation should be used only first time, then only acroyyms should be used in the entire paper.

- This paper need minor modifications. Below are my comments:

- Abstract can be reduced. The abstract is NOT satisfactory because it didn't contain the following parts:

i. The importance of or motivation for the research.

ii. The issue/argument of the research.

iii. The methodology.

iv. The result/findings.

v. The implications of the result/findings.

- In the first four paragraphs of literature review section, the authors have presented a good references, but they need to present the recent and most updated references

- Authos should discuss the dataset extensively.

- The quality of the figures can be improved more. Figures should be eye-catching. It will enhance the interest of the reader.

- Please highlight the contribution clearly in the introduction

- Some Paragraphs in the paper can be merged and some long paragraphs can be split into two.

- The discussion is very important in research paper. Nevertheless, this section is short and should be presented completely.

-- Authors should add the most recent reference:

1) BCD-WERT: a novel approach for breast cancer detection using whale optimization based efficient features and extremely randomized tree algorithm, PeerJ Computer Science 7, e390

2) Classification of COVID-19 individuals using adaptive neuro-fuzzy inference system, Multimedia Systems, 1-15

Reviewer #2: In Introduction section, the drawbacks of each conventional technique should be described clearly.

The authors should emphasize the difference between other methods to clarify the position of this work further.

The motivation for the present research would be clearer, by providing a more direct link between the importance of choosing your own method.

Define all the variables before using

The writing of the paper needs a lot of improvement in terms of grammar, spellings, and presentations. The paper needs careful English polishing since there are many typos and poorly written sentences.

The authors can cite the following references

Antlion re-sampling based deep neural network model for classification of imbalanced multimodal stroke dataset

An AI-based intelligent system for healthcare analysis using Ridge-Adaline Stochastic Gradient Descent Classifier

6. PLOS authors have the option to publish the peer review history of their article (what does this mean?). If published, this will include your full peer review and any attached files.

Reviewer #1: No

Reviewer #2: No

---

## [Author Response · Author response to Decision Letter 0]

24 Apr 2021

Sir PLOS One’s Editor (s),

Hoping you are all well;

Special thanks for your decision letter on our manuscript No. PONE-D-21-09994.

Concerning the following hint from you:

 “We noted in your submission details that a portion of your manuscript may have been presented or published elsewhere. „Our submitted manuscript intitled: “Automatic Semantic Segmentation of Breast Tumors in Ultrasound Images Based on Combining Fuzzy Logic and Deep Learning – A Feasibility Study”, is a replacement to the submission No. PONE-D-21-07670 (Efficient Automatic Semantic Segmentation of Breast Tumors in Ultrasound Images Based on Combining Fuzzy Logic and Deep Learning). Please clarify whether this publication was peer-reviewed and formally published. If this work was previously peer-reviewed and published, in the cover letter please provide the reason that this work does not constitute dual publication and should be included in the current manuscript.” 

We assure that this paper has not been presented or published elsewhere. The issue of the old withdrawn submission “PONE-D-21-07670” is that more changes have been needed. So, we have withdrawn it and modified it totally, and then we have submitted it again to PLOS One as: PONE-D-21-09994. So, again we assure that our submission has not been presented or published elsewhere.

Concerning Reviewer requirements and PLOS one requirements: 

Kindly, fined the final modified manuscript (file name: “Manuscript.docx”). 

Also, find the two accompanied files: “Response to Reviewers.pdf” and “Revised Manuscript with Track Changes.pdf”.

Concerning Figures: All 15th Figs have been adapted by PACE (https://pacev2.apexcovantage.com/) – and been submitted individually. But, their space into Manuscript has been saved by stating each Fig’s legend. So, kindly, put each fig above its legend as stated in the final “Manuscript.docx” file.

Hoping the modifications can meet with both reviewers and PLOS One requirements.

Best Regards;

Mohammed Tarek GadAllah, Assistant Researcher (M. Electronic Eng.) with Computers and Systems Department, Electronics Research Institute (ERI), Joseph Tito St, Huckstep, El Nozha, Cairo, Egypt. 

E-mails: mohammed.tag.1986@eri.sci.eg; mohammed.tag.1986@gmail.com; mohamed_msc_1986@yahoo.com

Phone: 00201012151263

---

## [Decision Letter · Decision Letter 1]

6 May 2021

Automatic Semantic Segmentation of Breast Tumors in Ultrasound Images Based on Combining Fuzzy Logic and Deep Learning – A Feasibility Study

PONE-D-21-09994R1

Dear Dr. GadAllah,

We’re pleased to inform you that your manuscript has been judged scientifically suitable for publication and will be formally accepted for publication once it meets all outstanding technical requirements.

Kind regards,

Thippa Reddy Gadekallu

Academic Editor

PLOS ONE

Additional Editor Comments (optional):

Reviewers' comments:

Reviewer's Responses to Questions

**Comments to the Author**

1. If the authors have adequately addressed your comments raised in a previous round of review and you feel that this manuscript is now acceptable for publication, you may indicate that here to bypass the “Comments to the Author” section, enter your conflict of interest statement in the “Confidential to Editor” section, and submit your "Accept" recommendation.

Reviewer #1: All comments have been addressed

2. Is the manuscript technically sound, and do the data support the conclusions?

Reviewer #1: Yes

3. Has the statistical analysis been performed appropriately and rigorously? 

Reviewer #1: Yes

4. Have the authors made all data underlying the findings in their manuscript fully available?

Reviewer #1: Yes

5. Is the manuscript presented in an intelligible fashion and written in standard English?

Reviewer #1: Yes

6. Review Comments to the Author

Reviewer #1: The authors have addressed my suggestions. I would like to accept this paper.

The authors have addressed my suggestions. I would like to accept this paper.

7. PLOS authors have the option to publish the peer review history of their article (what does this mean?). If published, this will include your full peer review and any attached files.

Reviewer #1: No

---

## [Editor Report · Acceptance letter]

12 May 2021

PONE-D-21-09994R1 

Automatic Semantic Segmentation of Breast Tumors in Ultrasound Images Based on Combining Fuzzy Logic and Deep Learning – A Feasibility Study 

Dear Dr. GadAllah:

I'm pleased to inform you that your manuscript has been deemed suitable for publication in PLOS ONE. Congratulations! Your manuscript is now with our production department. 

Kind regards, 

on behalf of

Dr. Thippa Reddy Gadekallu 

Academic Editor

PLOS ONE